# In Silico Design of New Dual Inhibitors of SARS-CoV-2 M^PRO^ through Ligand- and Structure-Based Methods

**DOI:** 10.3390/ijms24098377

**Published:** 2023-05-06

**Authors:** Alessia Bono, Antonino Lauria, Gabriele La Monica, Federica Alamia, Francesco Mingoia, Annamaria Martorana

**Affiliations:** 1Dipartimento di Scienze e Tecnologie Biologiche, Chimiche e Farmaceutiche “STEBICEF”, University of Palermo, Viale delle Scienze, Ed. 17, 90128 Palermo, Italy; alessia.bono01@unipa.it (A.B.); gabriele.lamonica01@unipa.it (G.L.M.); annamaria.martorana@unipa.it (A.M.); 2Istituto per lo Studio dei Materiali Nanostrutturati (ISMN), Consiglio Nazionale delle Ricerche (CNR), Via Ugo La Malfa, 153, 90146 Palermo, Italy; francesco.mingoia@ismn.cnr.it

**Keywords:** catalytic site, allosteric site, SARS-CoV-2 M^PRO^, inhibitors, benzo[*b*]thiophene, benzo[*b*]furan, dual binding site activities

## Abstract

The viral main protease is one of the most attractive targets among all key enzymes involved in the life cycle of SARS-CoV-2. Considering its mechanism of action, both the catalytic and dimerization regions could represent crucial sites for modulating its activity. Dual-binding the SARS-CoV-2 main protease inhibitors could arrest the replication process of the virus by simultaneously preventing dimerization and proteolytic activity. To this aim, in the present work, we identified two series’ of small molecules with a significant affinity for SARS-CoV-2 M^PRO^, by a hybrid virtual screening protocol, combining ligand- and structure-based approaches with multivariate statistical analysis. The Biotarget Predictor Tool was used to filter a large in-house structural database and select a set of benzo[*b*]thiophene and benzo[*b*]furan derivatives. ADME properties were investigated, and induced fit docking studies were performed to confirm the DRUDIT prediction. Principal component analysis and docking protocol at the SARS-CoV-2 M^PRO^ dimerization site enable the identification of compounds **1b**,**c**,**i**,**l** and **2i**,**l** as promising drug molecules, showing favorable dual binding site affinity on SARS-CoV-2 M^PRO^.

## 1. Introduction

SARS-CoV-2 (Severe Acute Respiratory Syndrome CoronaVirus 2) is a highly transmissible and pathogenic coronavirus and represents the etiologic agent of COronaVIrus Disease-19 (COVID-19) [1,2]. The β-coronavirus SARS-CoV-2 belongs to the family of enveloped viruses with a positive single-stranded RNA genome (+ssRNA) [3,4], which also includes the previously identified viruses SARS-CoV (2002) and MERS-CoV (Middle East Respiratory Syndrome CoronaVirus 2013), both of which share a large part of genome with SARS-CoV-2 [5,6].

In an outbreak scenario, caused by the rapid global spread of SARS-CoV-2, the worldwide research community points to the urgency of developing an effective therapeutic strategy to support vaccination campaigns [7]. In this light, the “drug repurposing” approach represented a successful strategy, allowing to bypass the lengthy process of pharmacokinetic and toxicological clinical trials required for the approval of any new drugs [8,9,10,11].

Several viral proteins were investigated as possible SARS-CoV-2 druggable targets, with the Main Protease (M^PRO^) emerging as one of the most attractive proteins [12,13].

The SARS-CoV-2 M^PRO^, also known as 3C-like protease (3CL^PRO^), is a cysteine catalytic enzyme with a pivotal role in mediating viral replication and transcription. In particular, the viral polyproteins, which are released from translated RNA, are processed by the SARS-CoV-2 M^PRO^ at 11 conserved sites generating 12 fragments (non-structural proteins, NPSs) [14]. The M^PRO^ is characterized by the dimerization of two monomers (A and B). Each monomer consists of three domains connected by long loop regions. Domain I (8–101) comprises six β-strands and one α-helix, while domain II (102–184) and domain III (201–303) include six β-strands and five α-helices, respectively (Figure 1). The binding pocket is divided into four subsites, S1′, S1, S2, and S3/S4: S1′ represents the catalytic site and is located between domains I and II, involving the Cys^145^/His^41^ catalytic dyad; S1 is characterized by the side chains of Phe^140^, Asn^142^, His^163^, His^164^, Glu^166^, and His^172^; S2 consists of hydrophobic amino acids, such as Met^49^, Tyr^54^, Met^165^, Pro^168^, Val^186^; finally S3/S4, including residues Gln^189^, Ala^191^, Gln^192^, Gly^251^, is particularly exposed to the solvent [15].

According to the nomenclature introduced by Schechter and Berger, the amino acids of the viral polyproteins processed by proteases are indicated as –P3–P2–P1↓P1′–, where P2 tolerates more hydrophobic amino acid with a clear preference for leucine, P1′ is a non-conserved prime recognition site (a small amino acid, such as serine, glycine, or alanine), P1 is a glutamine, and the arrow ↓ represents the cleavage site between P1 and P1′ (Figure 2) [13,17]. The M^pro^ exclusively cleaves polypeptide sequences after a glutamine residue [18]. Since no human host-cell proteases are known with this substrate specificity, the main protease is a selective biological target for COVID-19 antiviral treatment.

A detailed structural analysis of the M^PRO^ protein revealed the presence of a potential allosteric site between the domains II and III, which is involved in the protease dimerization, Figure 1 [19,20,21]. In particular, the *N*-finger residues Ser^1^-Gly^11^, residue Asn^214^, the region around residues Glu^288^, Asp^289^, Glu^290^ in a tight contact with the *N*-finger, and the C-terminal last helix region around residues Arg^298^, Gln^299^ are involved in the hydrogen bond interactions between the two SARS-CoV-2 M^PRO^ protomers (Arg^4^/Glu^290^, Gly^11^/Glu^14^, Ser^1^/Glu^166^, Ser^301^/Ser^139^, Thr^304^/Glu^166^, Ser^123^/Arg^298^, Ser^139^/Gln^299^, Arg^4^/Gln^299^) [22]. The dimerization of the protein is a necessary step for the catalytic activity, because in the monomeric state the active site pocket collapses and is not available for the binding with the substrate.

A lot of examples of ligand/structure-based, hybrid or nonhybrid, virtual screenings, that led to biologically interesting compounds, have been reported in the literature to date.

QSAR and/or pharmacophore modelling approaches combined with molecular docking, MD simulations and free binding energy MM/PBSA allowed for the design of new derivatives with promising SARS-CoV-2 M^PRO^ inhibition activity [23,24,25,26,27,28,29,30,31,32].

In addition to the common techniques, quantum mechanics/molecular mechanics (QM/MM) calculations, although less common, are beginning to emerge as reliable computational approaches in the design of covalent inhibitors, elucidating the mechanisms, kinetics, and thermodynamics of covalent modification of a target protein [33,34]. Furthermore, from this point of view, it should be evidenced as the integration of non-covalent and covalent docking protocols leading to compounds with optimal interaction both with the catalytic residue and to the other clefts in the binding site [25,35,36]. In fact, in the past three years, a great number of Structure-Based Virtual Screenings (SBVSs) facilitated the identification of some efficacious SARS-CoV-2 M^PRO^ covalent and/or noncovalent inhibitors; overall, this was possible thankfully to the large database of solved SARS-CoV-2 M^PRO^ crystallographic structures deposited in the Protein Data Bank (PDB) [25,27,30,31].

Additionally, innovative techniques were developed, such as the combined protocol of Advanced Deep Q-learning Network and Fragment-Based Drug Design (ADQN-FBDD), an artificial intelligence (AI), with structure-based drug design (SBDD) [37,38,39].

The importance of SARS-CoV-2 M^PRO^ in viral replication and transcription, as well as its absence in human cells, makes M^PRO^ a potent target for the design of antiviral drugs [12,13,14]. In particular, the presence of two binding sites, the catalytic and the allosteric one, has led us to focus our research on the development of selective small molecules, with inhibition activity both on the SARS-CoV-2 M^PRO^ catalytic binding site and the dimerization one. Targeting dimerization could potentially affect the substrate pocket and thus inhibit the M^PRO^ activity in the sense of allosteric non-competitive inhibition [40,41]. Indeed, this approach allows, in a first phase, to inhibit the activation of the protein by preventing its dimerization and, in a second moment, when the protein is already in a dimeric active form, to directly block the catalytic activity.

## 2. Results and Discussion

Virtual screening is a reliable approach in the search for candidates able to target SARS-CoV-2. Among all the structural/non-structural viral proteins of SARS-CoV-2, M^PRO^ proved to be the most attractive target for the development of selective antiviral drugs. In the present work, we propose in silico studies to analyze the SARS-CoV-2 M^PRO^ inhibition activity of small molecules with the aim of identifying new promising dual binding site modulators of SARS-CoV-2 M^PRO^.

Figure 3 shows the flowchart of the in silico mixed ligand-structure screening for the identification of new SARS-CoV-2 M^PRO^ inhibitors. The computational protocol was based on a hierarchical workflow with an initial phase of filtering using a ligand-based approach focused on DRUDIT^ONLINE^ (DRUg DIscovery Tools, open access web-service, www.drudit.com, accessed on 20 March 2023) [42], a free online resource based on the calculation of molecular descriptors by MOLDESTO (MOLecular DEScriptors TOol) [42]. The use of DRUDIT Biotarget Predictor Tool (BPT) allowed the selection of two clusters of small molecules already known in the literature for their antitumor activity [43,44] and characterized by a benzo[*b*]thiophene or a benzo[*b*]furan scaffold. Absorption, Distribution, Metabolism, and Excretion (ADME) properties were examined using the SwissADME web tool, a robust predictive model to support biocompatible drug discovery projects [45].

Then, to improve the in silico affinity results, the 24 selected compounds were investigated by structure-based molecular docking studies at the catalytic binding site.

The matrix of molecular descriptors for the 24 molecules, obtained through the DRUDIT web-service, was merged with the sequence of molecular descriptors of the non-covalent allosteric inhibitor pelitinib, to perform a multivariate analysis. Finally, the resulting 13 benzothiophene and benzofuran compounds were analyzed using structure-based molecular docking protocols at the dimerization binding site.

### 2.1. In Silico Ligand-Based Approach: DRUDIT^ONLINE^

To perform the in silico ligand-based approach, we used the DRUDIT Biotarget Predictor Tool (BPT), which allows the prediction of the affinity of input structures to a selected biological target [42].

The first step required the construction of a template for the SARS-CoV-2 M^PRO^ binding site, following the method recently described in the literature [46]. Many well-known drugs were used to perform molecular docking studies and evaluate their ability to bind to the catalytic site of the SARS-CoV-2 M^PRO^. The resulting best scored molecules were used to build the SARS-CoV-2 M^PRO^ template, which was implemented in DRUDIT. Subsequently, an in-house structure database of approximately 10,000 heterocyclic structures was uploaded to the web-service DRUDIT PBT. Standard parameters (N = 500, Z = 50, G = a) [46] were used, and the output data were ranked according to the Drudit Affinity Score (DAS), a parameter whose value (in the range 0/1, low/high affinity), reflects the capability of compounds to bind into the SARS-CoV-2 M^PRO^ catalytic site.

The analysis of the results, applying the cut-off value of 0.8 to DAS, allowed to identify ethyl 3-benzoylamino-5-[(1*H*-imidazol-4-yl-methyl)-amino]-benzo[*b*]thiophene-2-carboxylate and ethyl 3-benzoylamino-5-[(1*H*-imidazol-4-yl-methyl)-amino]-benzo[*b*]furan-2-carboxylate compounds **1** and **2** [43,44] (Figure 4) as interesting heterocyclic small molecules for the inhibition of SARS-CoV-2 M^PRO^ via modulation of the catalytic active site. Table 1 shows the DAS values of the 24 selected molecules, characterized by a central heterocyclic benzo[*b*]thiophene or benzo[*b*]furan core and two side moieties: the substituted 3-benzoylamino and the imidazole one.

### 2.2. ADME Properties

The 24 selected molecules were submitted to the SwissADME web-tools (http://www.swissadme.ch, accessed on 20 March 2023) [45] considering a set of well consolidated parameters for searching bioactive compounds, such as PAINS filters [47], Lipinski’s rules [48], Veber [49], and Egan filters [50]. The analysis of the data showed in Table 2 highlighted that the benzo[*b*]thiophene and benzo[*b*]furan compounds generally met the expectations in terms of bioactivity. Thirteen of the twenty-four structures have no violations, and all compounds have no PAINS. In light of these considerations, no compounds were excluded for the in silico structure-based analysis.

All the parameters calculated through SwissADME are present in the Appendix A.

### 2.3. In Silico Structure-Based Studies: Molecular Docking at the Catalytic Site of SARS-CoV-2 M^PRO^

Induced Fit Docking (IFD) studies were performed to validate the obtained ligand-based data and to gain insight into the structural features of ligand/SARS-CoV-2 M^PRO^ (pdb code 7VH8 [51]) complexes, analyzing the mutual conformational changes between ligands and proteins.

We focused the docking grid on the SARS-CoV-2 M^PRO^ binding pocket, including the four subsites S1′, S1, S2, S3/S4 as described in the Section 3. Figure 5b shows the 3D active binding site of SARS-CoV-2 M^PRO^ in covalently bonding with nirmatrelvir (PF-07321332, 2D structure in Figure 5a), a second-generation orally available protease inhibitor currently in phase three clinical trials in combination with ritonavir (PAXLOVID^®^, see ClinicalTrials.gov identifier: NCT04960202).

The IFD studies aim to confirm the DRUDIT prediction and the capability of ethyl 3-benzoylamino-5-[(1*H*-imidazol-4-yl-methyl)-amino]-benzo[*b*]thiophene-2-carboxylates of type **1** and ethyl 3-benzoylamino-5-[(1*H*-imidazol-4-yl-methyl)-amino]-benzo[*b*]furan-2-carboxylates of type **2** to effectively interact with the selected target binding site. Table 3 shows the IFD and docking scores of the selected 24 structures **1a**–**l** and **2a**–**l**, and the reference ligand nirmatrelvir (PF-07321332) [51].

The IFD analysis confirmed the biotarget affinity results and identified the benzo[*b*]thiophenes **1a**–**d**,**f**,**i**–**l** and benzo[*b*]furans **2h**,**i**,**l** as the most promising competitive inhibitors (IFD score range from −675.768 to −673.730) of the SARS-CoV-2 M^PRO^ catalytic binding site, with higher IFD scores than nirmatrelvir (IFD score −673.142) (Table 3).

Table 4 provides the overview of the amino acids involved in the binding with the 12 highest scoring compounds. The labelled residues were highlighted by the analysis of 2D and 3D ligand pose maps at a distance of 3 Å.

Most of the benzo[*b*]thiophenes **1a**–**d**,**f**,**i**–**l** and benzo[*b*]furans **2h**,**i**,**l** interact with amino acids Thr^25^, Thr^26^, Leu^27^, Met^49^, Phe^140^, Leu^141^, Asn^142^, Gly^143^, Cys^145^, His^163^, Met^165^, Glu^166^, Pro^168^, Arg^188^, Gln^189^, Thr^190^, Gln^192^ with the same reversible interactions observed for the reference compound nirmatrelvir. In addition, compounds **1a**–**d**,**f**,**i**,**k**,**l** and **2h**,**i**,**l** establish more interactions than nirmatrelvir with the conserved amino acids of the four SARS-CoV-2 M^PRO^ sub-regions S1, S1′, S2, and S3/S4, suggesting an improved affinity of the benzo[*b*]thiophene and benzo[*b*]furan compounds for the catalytic binding site and resulting in more stable ligand/protein complexes.

As shown in Figure 6b,d, the binding pocket exhibits suitable properties for ligands **1d** and **2l**, which are the two highest IFD scoring compounds (2D structures shown in Figure 6a,c). Both compounds interact with the SARS-CoV-2 M^PRO^ binding site by extending all substituents into the four subsites S1′, S1, S2, S3/S4, and creating a network of key reversible hydrogen bonds with the amino acids: His^41^, Phe^140^, Asn^142^, Cys^145^, Glu^166^, Gln^189^, Thr^190^, Tyr^54^, Met^165^, Val^186^ and His^41^, Phe^140,^ Asn^142^, Cys^145^, Glu^166^, Gln^189^Thr^190^, Tyr^54^, Met^165^, Val^18^ for benzo[*b*]thiophene **1d** and benzo[*b*]furan **2l**, respectively.

Among the ligands with higher IFD scores, the analysis of the binding poses of derivatives **1b**, **1c**, **1l** and **2l**, shown in Figure 7, highlights a remarkable overlap of poses, indicating a redundancy in the position of the key elements of the small molecules within the four sub-pockets, and forming a large number of interactions. The imidazole moiety, charged at physiological pH, is capable to penetrate deeply into the S1 and S3/S4 sub-regions (Asn^142^, Gly^143^, His^163^, His^164^, Glu^166^) and stabilize itself by forming hydrogen interactions with the side chains and/or the backbone of residues, that are particularly exposed to the solvent in the S3/S4 site. Probably, this portion could mimic the Gln residue of the natural substrates, similarly to the five membered γ-lactamic ring, which is the most recurrent fragment of selective inhibitors at the catalytic binding site, reported in the literature [13].

The SARS-CoV-2 M^PRO^ S2 cleft, comprised mainly of hydrophobic amino acids (Met^49^, Tyr^54^, Met^165^, Leu^167^, Pro^168^, Val^186^, Asp^187^, Arg^188^), appears to be very flexible, allowing it to bind both small and bulky aromatic/alkyl portions. The heterocyclic scaffold could represent a central pharmacophoric portion, thanks to its capability to stabilize the ligand/protein complex in favorable ligand positions. Additional π-π staking interactions are observed between the benzo[*b*]thiophene and benzo[*b*]furan ring systems and the imidazole substituent of His^41^.

The carboxyethyl moiety is stabilized in the S1′ pocket, instead of the carboxyamide moiety, simulating a peptide bond of natural substrate which creates several interactions with the side chains of His^41^, Asn^142^ and Glu^166^, in both S1′ and S1 pockets. The substituted phenyl rings are arranged in the region adjacent to the S1 cleft.

### 2.4. Statistical Analysis: Principal Component Analysis (PCA)

The structural features of the selected compounds of types **1** and **2** prompted us to evaluate them also as potential binders for the allosteric dimerization site of the SARS-CoV-2 M^PRO^, as non-competitive inhibitors. Similarly to the known dimerization site inhibitor pelitinib, the benzo[*b*]thiophene and benzo[*b*]furan compounds have a series of aromatic rings, linked by rotatable bonds.

To investigate the potential inhibitory activity of the selected compounds at the dimerization site of SARS-CoV-2 M^PRO^, we performed a Principal Component Analysis (PCA), including the molecular descriptor matrix of the 24 compounds, obtained from DRUDIT ligand-based studies, and the molecular descriptors of the non-covalent allosteric inhibitor pelitinib.

The application of PCA to the matrix of structures versus molecular descriptors (Appendix A), showed a total variance of 50% expressed by the first two components. The bidimensional plot (PC1 versus PC2, Figure 8) shows the 2D arrangement of the molecules in the graph compared with the reference ligand pelitinib.

Based on the distribution of the compounds in this graph, the molecules in the proximity of pelitinib were identified as new inhibitors potentially capable of modulating the dimerization process. The distances of the molecules from the reference pelitinib were calculated and the data were listed in Table 5, which provides cartesian coordinates for the derivatives in the circle (Figure 8), (see Appendix A, for all coordinates). Benzo[*b*]thiophenes **1b**,**c**,**g**–**i**,**l** and benzo[*b*]furans **2a**–**c**,**g**–**i**,**l** were selected to be further investigated with structure-based studies, considering the dimerization region of SARS-CoV-2 M^PRO^ as allosteric binding site.

### 2.5. In Silico Structure-Based Studies: Induced Fit Docking (IFD) into the Allosteric Site of SARS-CoV-2 M^PRO^

Following the statistical analyses, IFD simulations performed by fixing the docking grid on the SARS-CoV-2 M^PRO^ allosteric site on pdb code 7AXM [41] were performed. In Figure 9b, the crystallographic structure of SARS-CoV-2 M^PRO^ complexed with non-covalent allosteric inhibitor pelitinib is shown (2D structure in Figure 9a).

The analysis of the results shows compounds with higher affinity than pelitinib. In Table 6 the IFD and docking values of the selected derivatives and of the reference compound are reported. In particular, the benzo[*b*]thiophenes **1b**,**c**,**g**,**i**,**l** and benzo[*b*]furans **2b**,**i**,**l** exhibit IFD scores in the range from −675.768 to −673.730, suggesting an interesting allosteric affinity for the dimerization binding site of the SARS-CoV-2 M^PRO^.

Considering the IFD results for both catalytic and dimerization sites, compounds **1b**, **c**,**i**,**l** and **2i**,**l** were found to have interesting IFD scores, suggesting a dual binding site inhibitory activity of SARS-CoV-2 M^PRO^.

Analysis of the amino acid maps (Table 7), in combination with the 2D and 3D poses examination of the ligands on the dimerization site uncovered promising expectations. Most benzo[*b*]thiophenes **1b**,**c**,**i**,**l** and benzo[*b*]furans **2i**,**l** interact with amino acids Thr^154^, Pro^252^, Gln^256^, Val^297^, Arg^298^, Cys^300^, Val^303^, Thr^304^, with similar reversible interactions observed with the reference compound pelitinib.

The selected compounds establish hydrogen bonds with both the residues of the N-finger region and the region around residues Arg^298^, Cys^300^, Ser^301^. The allosteric ligand/protein complex in the pocket between the domains II and III could interfere with the necessary interactions between the two monomers and prevent the dimerization process with the consequent inactivation of the SARS-CoV-2 M^PRO^.

In Figure 10, compounds **1c** and **1l** are shown in a recurring position, fitting in the SARS-CoV-2 M^PRO^ allosteric site (dimerization domain), in which there are highly conserved with essential amino acids, such as *N*-finger residues (Ser^1^, Gly^2^, Phe^3^, Arg^4^, Lys^5^, Met^6^, Ala^7^), Pro^293^, Phe^294^, Asp^295^, Val^296^, Val^297^, Arg^298^, Gln^299^, Cys^300^, Ser^301^.

### 2.6. Molecular Dynamic Simulation

Molecular dynamic simulations were performed to investigate the stability and dynamics of the SARS-CoV-2 M^PRO^ in complex with **1d** (catalytic site, 7VH8) and **1c** (allosteric site, 7AXM). The response was studied in terms of protein and ligand binding energy, demonstrating high stability across the simulation time (15 and 20 ns for **1d** and **1c**, respectively) and reaching a plateau energy (time-energy graphs for both complexes are available in Appendix A). This further analysis allowed us to confirm the robustness of the in silico protocol.

## 3. Materials and Methods

### 3.1. Ligand-Based Studies

The web-service DRUDIT 1.0 (www.drudit.com, accessed on 20 March 2023) operates through four servers, each of which can perform more than ten jobs simultaneously, and several software modules implemented in C and JAVA running on MacOS Mojave. The Biotarget Finder Module was used to screen small molecule drug candidates as SARS-CoV-2 M^PRO^ inhibitors [42].

#### Biotarget Predictor Tool (BPT)

The tool provides prediction of the binding affinity between candidate molecules and the specified biological target. The template of SARS-CoV-2 M^PRO^ was created using a set of well-known drugs to perform molecular docking studies at catalytic site of SARS-CoV-2 M^PRO^. It was uploaded in DRUDIT, and the default DRUDIT parameters (N = 500, Z = 50, G = a) were used [42,46]. In accordance with the first phase of the in silico workflow, the database was uploaded in DRUDIT and submitted to the Biotarget Predictor. The output results were obtained as DAS (Drudit Affinity Scores) for each structure, reflecting the binding affinity of compounds against the SARS-CoV-2 M^PRO^ catalytic site.

### 3.2. Structure-Based Studies

The preparation process of ligands and protein-ligand complexes used for in silico studies was performed as detailed below:

#### 3.2.1. Ligand Preparation

The ligands for docking were prepared through the LigPrep tool, available in the Maestro Suite 2022, Schrödinger software [52]. For each ligand, all possible tautomers and stereoisomers were generated for a pH of 7.0 ± 0.4, using default setting, through the Epik ionization method [53]. Consequently, the integrated Optimized Potentials for Liquid Simulations (OPLS) 2005 force field was used to minimize the energy status of the ligands [54].

#### 3.2.2. Protein Preparations

The two crystal structures of SARS-CoV-2 M^PRO^ (pdb codes 7VH8 [51], 7AXM [41]) were downloaded from the Protein Data Bank [55,56]. As regard the pdb code 7VH8, a first breakup of the covalent bound between the cocrystal ligand and the Cys^145^ was carried out.

Successively, both the two protein structures were prepared using the Protein Preparation Wizard, in the Schrödinger software, with the default setting [57]. In detail, bond orders were assigned, including Het group, hydrogen atoms were added, all water molecules were delated, and protonation of the heteroatom states were carried out using the Epik-tool (with the pH set at biologically relevant values, i.e., at 7.0 ± 0.4). The H-bond network was then optimized. The structure was finally subjected to a restrained energy minimization step (RMSD of the atom displacement for terminating the minimization was 0.3 Å), using the OPLS 2005 force field [54].

#### 3.2.3. Docking Validation

Molecular docking studies were executed and scored by using the Glide module, available in the Schrödinger Suite program package. The receptor grids were obtained through assignment the original ligands (PF-07321332 and pelitinib for pdb codes 7VH8 [51] and 7AXM [41], respectively) as the centroid of the grid boxes. Extra Precision (XP) mode, as scoring function, was used to dock the generated 3D conformers into the receptor model. The post-docking minimization step was performed with a total of five poses for each ligand conformer, and a maximum of two docking poses were generated per ligand conformer. The proposed docking procedure was able to re-dock the original ligands within the receptor-binding pockets with RMSD   0.51 Å. Table 8 shows all the parameters combinations used for RMDS value optimization for both the proteins. In detail, radii Van der Waals scaling allows us to temporarily remove active-site residue side chains. By default, the scaling factor is 0.50 for the receptor and 0.50 for the ligand, with a partial charge threshold of 0.15. Removing the side chains from active site residues provides more room for ligand docking, so the receptor does not need to be quite as soft. The side chains are restored after docking.

The side chain optimization makes possible to reduce the distance from the ligand that defines residues for refinement. In general, the optimal value for this parameter is set to 5.0 Å by default, ensuring that the optimal setting for side chains is selected.

The energy minimization parameters controls the minimization protocol through the distance-dependent dielectric constant (the optimum is to set the protein/ligand dielectric constants to values of 1–2, default setting at 2) and the maximum number of minimization steps (by default fixed to 100) [54].

#### 3.2.4. Induced Fit Docking

Induced fit docking simulation was performed using the IFD application, an accurate and robust Schrödinger technology that accounts for both ligand and receptor flexibility [58,59]. Schrödinger’s induced fit docking validated protocol was applied by using the SARS-CoV-2 M^PRO^ protein from the PDB (pdb codes 7VH8 [51] and 7AXM [41]), previously refined by the Protein Preparation module. The IFD score (IFD score = 1.0 Glide Gscore + 0.05 Prime Energy), which includes protein–ligand interaction energy and system total energy, was calculated and used to rank the IFD poses. The more positive in modulus the IFD score, the more favorable the binding.

#### 3.2.5. Molecular Dynamic Simulation

Molecular dynamics simulations were performed using the MacroModel task available in Maestro Schrodinger for the top two best low binding energy ligand-protein complexes after docking: SARS-CoV-2 M^PRO^ (catalytic site, 7VH8) with **1d** and SARS-CoV-2 M^PRO^ (allosteric site, 7AXM) with **1c**. The OPLS-2005 force field was used to model the proteins and ligands and the systems were energy minimized for 1000 steps before a production run of 15 and 20 ns, respectively. The systems temperature was maintained around 300 K. The results were analyzed in terms of protein and ligand time-lapse binding energy.

### 3.3. Principal Component Analysis

PCA, one of the most-widely used multivariate exploratory techniques, enables a drastic dimensionality reduction of an original raw data, transforming the original matrix to a new one, whose set of variables, termed as Principal Components (PCs), appear to be ordered with descending importance in terms of variance. Principal Components Analysis can be highly useful for data classification and pattern recognition. In this work, DRUDIT was used to obtain the original matrix of objects versus variables (Appendix A), and free version TIBCO Statistica^®^ software was used to perform principal component analysis.

Grubb’s test, also known as ESD method (extreme studentized deviate), was performed. In detail, the outliers were determined by singularly evaluating those compounds outside the red circle (Figure 8) in comparison with the cluster of molecules closest to pelitinib. The identified outliers were not included in the next step of the virtual screening.

## 4. Conclusions

The SARS-CoV-2 M^PRO^ is a key cysteinyl enzyme for virus replication. Considering its mechanism of action, which consists of a preliminary dimerization with consequent activation of its catalytic site, small molecules capable of binding and interfering with both the dimerization and the proteolytic process could represent an interesting and more efficacious class of therapeutics, with a dual binding activity. With this aim, in silico approaches are reliable strategies in the search for candidates in drug discovery projects.

Here we identified a set of ethyl 3-benzoylamino-5-[(1*H*-imidazol-4-yl-methyl)-amino]-benzo[*b*]thiophene-2-carboxylates 1 and ethyl 3-benzoylamino-5-[(1*H*-imidazol-4-yl-methyl)-amino]-benzo[*b*]furan-2-carboxylates 2 as potential SARS-CoV-2 M^PRO^ inhibitors, through a hierarchical and hybrid virtual screening. In detail, the Biotarget Predictor ligand-based Tool, available in the open-access web-platform DRUDIT^ONLINE^ (www.drudit.com, accessed on 20 March 2023) allows for the filtering of a large in-house structure database, identifying the aforementioned set of small molecules with high affinity against the SARS-CoV-2 M^PRO^ catalytic binding site. ADME properties of the selected compounds were investigated through the SwissADME tool (www.swissadme.ch, accessed on 20 March 2023) and induced fit docking studies were performed on the catalytic site to confirm DRUDIT prediction. Moreover, aiming at evaluating the possibility of a dual binding mechanism of action, the identified hits were further investigated by means of principal component analysis and IFD into the dimerization site. By combining these results, compounds **1b**,**c**,**i**,**l** and **2i**,**l**, with favorable ADME properties (drug-likeliness, lead-likeliness, no PAINS), showed the capability to strongly bind both to the catalytic and allosteric SARS-CoV-2 M^PRO^ sites suggesting a potential dual activity. In general, considering the high amino acids conservation rate in both pockets, the potential drugs proposed here might even be effective against mutation variants and other coronaviruses.

In silico simulations of a complex system are becoming increasingly popular in the drug development process and across clinical research. The use of computational models and simulations offers significant advantages over human-based clinical trials in both operational factors and therapeutic outcomes, providing the tools to evaluate various treatments qualitatively and quantitatively on specific diseases, and offering more practical and economical experiments.

Even if multiple in silico virtual screenings on the SARS-CoV-2 M^PRO^ have been reported in literature in the past three years, our computational workflow is peculiar. In particular, we took the advantages of our in-house ligand based Biotarget Predictor Tool (BPT) which allowed us to screen an enormous ligands library in negligible computational time and with no need for particularly high-performance hardware. This tool, integrated with both structure-based techniques (molecular docking and dynamics) and, interestingly, multivariate statistical analysis, was applied to evaluate a potential dual binding activity on SARS-CoV-2 M^PRO^, which has been rarely explored in other virtual screening research. Furthermore, to the best of our knowledge, this is one of the first examples of virtual screenings focused on the identification of dual SARS-CoV-2 M^PRO^ inhibitors.

This analysis improves our knowledge of protein–ligand relationships in SARS-CoV-2 M^PRO^ catalytic and allosteric sites and offers new molecular scaffolds for inhibitor design.

## Figures and Tables

**Figure 1 ijms-24-08377-f001:**
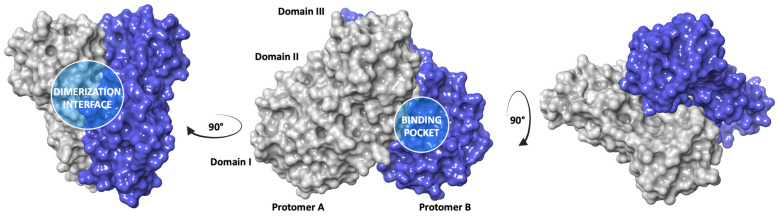
X-ray structure of dimeric SARS-CoV-2 M^PRO^ (pdb code 6Y2F) [16]; the two monomers are highlighted in blue and grey.

**Figure 2 ijms-24-08377-f002:**
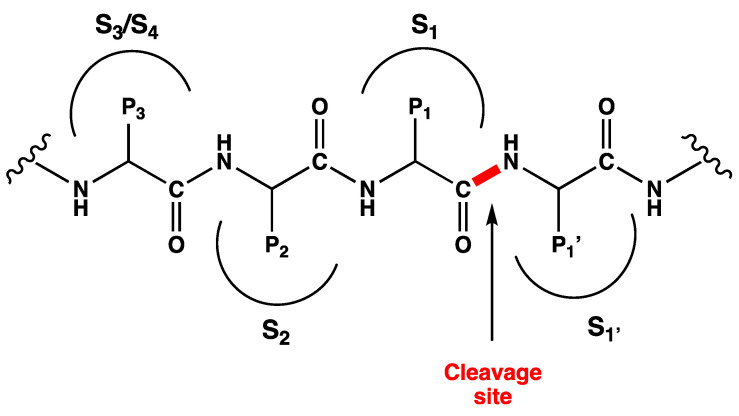
SARS-CoV-2 M^PRO^ viral polyproteins. Sub-regions of the binding pocket (S1′, S1, S2 and S3/S4) are labeled with S numbering complementary to the P fragments (P1′, P1, P2 and P3) of the viral polyproteins.

**Figure 3 ijms-24-08377-f003:**
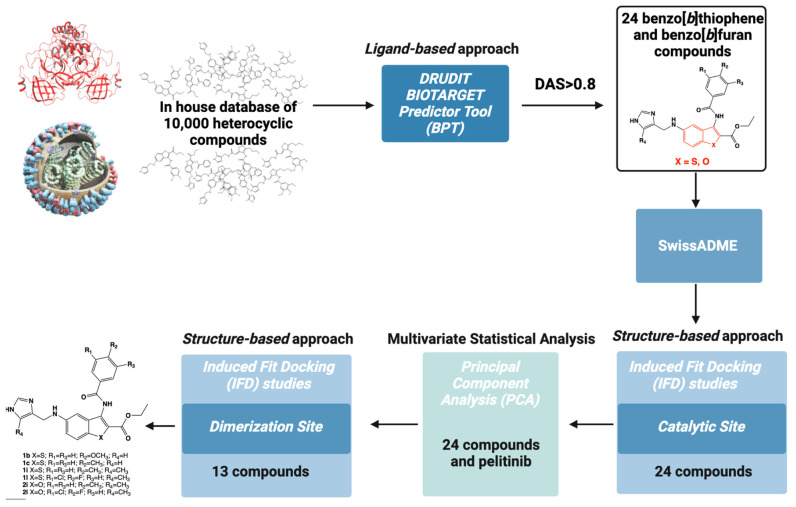
In silico protocol workflow for the identification of new SARS-CoV-2 M^PRO^ inhibitors as effective antiviral molecules in COVID-19 treatment.

**Figure 4 ijms-24-08377-f004:**
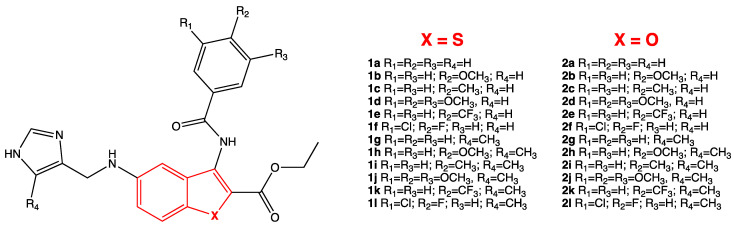
General structure of ethyl 3-benzoylamino-5-[(1*H*-imidazol-4-yl-methyl)-amino]-benzo[*b*]thiophene-2-carboxylates **1a**–**l** and ethyl 3-benzoylamino-5-[(1*H*-imidazol-4-yl-methyl)-amino]-benzo[*b*]furan-2-carboxylates **2a**–**l** as new potential antiviral molecules [43,44].

**Figure 5 ijms-24-08377-f005:**
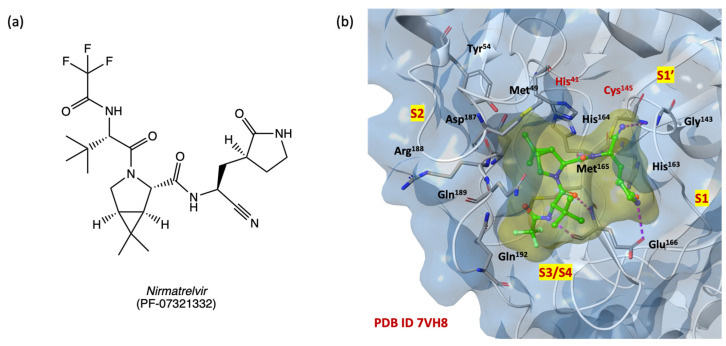
(**a**) Two-dimensional structure of nirmatrelvir; (**b**) SARS-CoV-2 M^PRO^ 3D binding site surface in complex with PF-07321332 (pdb code 7VH8) [51]; nitrogen, oxygen, sulfur, and fluorine atoms are in blue, red, yellow and light green, respectively.

**Figure 6 ijms-24-08377-f006:**
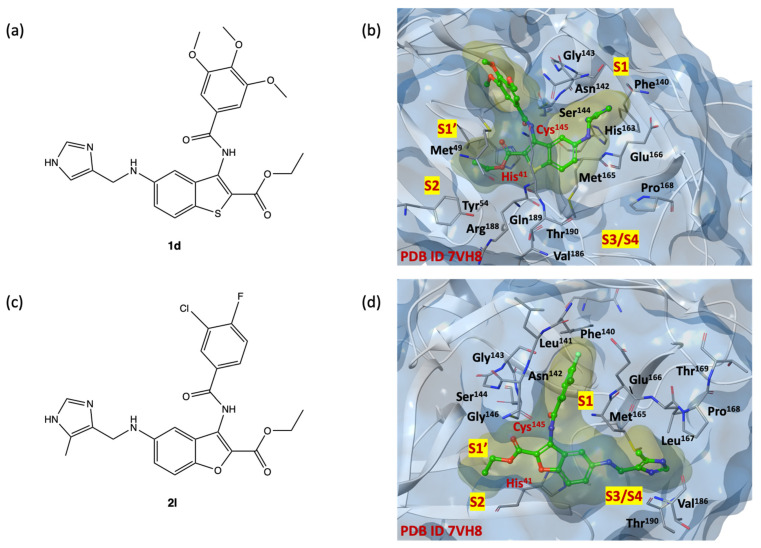
(**a**) Two-dimensional structure of benzo[*b*]thiophene **1d**; (**b**) 3D complex of SARS-CoV-2 M^PRO^ binding site (pdb code 7VH8) with benzo[*b*]thiophene **1d** [51]; nitrogen, oxygen, and sulfur atoms are in blue, red, and yellow, respectively. (**c**) 2D structure of benzo[*b*]furan **2l**; (**d**) 3D complex of SARS-CoV-2 M^PRO^ binding site (pdb code 7VH8) with benzo[*b*]furan **2l** [51]**;** nitrogen, oxygen, fluorine and chlorine atoms are in blue, red, light green and dark green, respectively.

**Figure 7 ijms-24-08377-f007:**
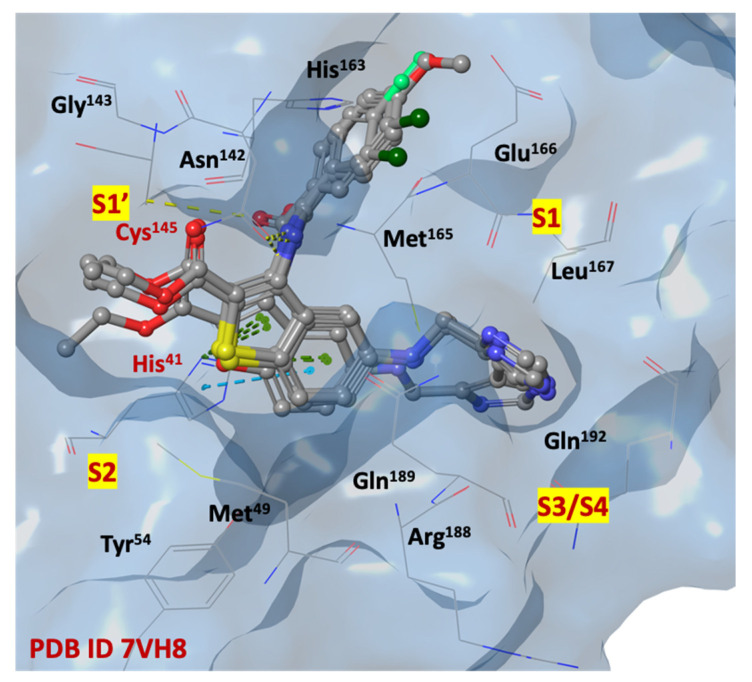
Three-dimensional overlaps of compounds **1b**, **1c**, **1l** and **2l** at SARS-CoV-2 M^PRO^ binding site (pdb code 7VH8) [51]; nitrogen, oxygen, fluorine, chlorine, and sulfur atoms are in blue, red, light green, dark green, and yellow, respectively.

**Figure 8 ijms-24-08377-f008:**
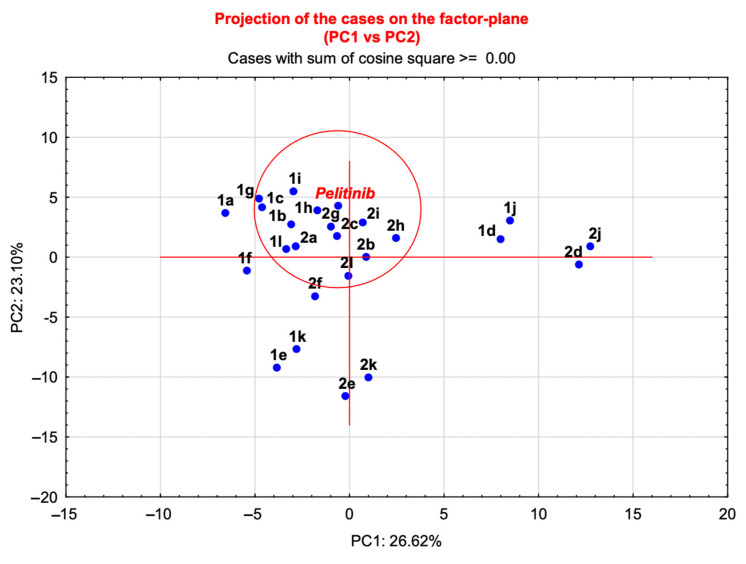
Principal Component Analysis (PC1 versus PC2) applied to the molecular descriptors matrix of the selected compounds and pelitinib.

**Figure 9 ijms-24-08377-f009:**
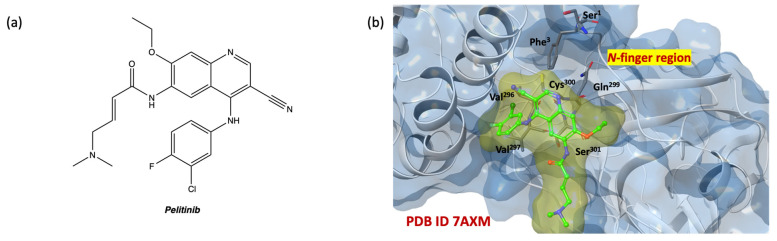
(**a**) Two-dimensional structure of pelitinib; (**b**) 3D complex of SARS-CoV-2 M^PRO^ allosteric site with pelitinib (pdb code 7AXM) [41]; nitrogen, oxygen, fluorine, chlorine, and sulfur atoms are in blue, red, light green, dark green, and yellow, respectively.

**Figure 10 ijms-24-08377-f010:**
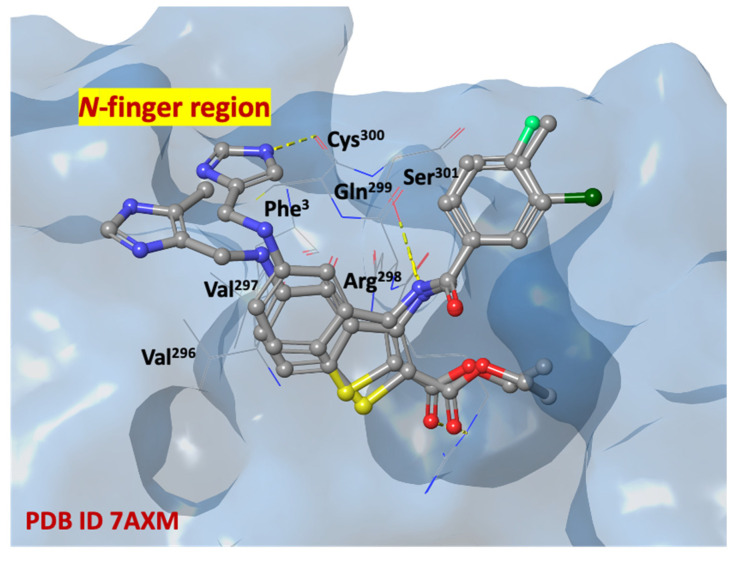
Three-dimensional overlaps of compounds **1c**, and **1l** at the SARS-CoV-2 M^PRO^ allosteric site (pdb code 7AXM) [41]; nitrogen, oxygen, fluorine, chlorine, and sulfur atoms are in blue, red, light green, dark green, and yellow, respectively.

**Table 1 ijms-24-08377-t001:** BIOTARGET Drudit Affinity Score (DAS) for benzo[*b*]thiophene and benzo[*b*]furan compounds **1a**–**l** and **2a**–**l**.

Compound	X	R_1_	R_2_	R_3_	R_4_	DAS Score
**2b**	O	H	OCH_3_	H	H	0.940
**2a**	O	H	H	H	H	0.922
**2c**	O	H	CH_3_	H	H	0.922
**1b**	S	H	OCH_3_	H	H	0.918
**2g**	O	H	H	H	CH_3_	0.916
**2f**	O	Cl	F	H	H	0.910
**2l**	O	Cl	F	H	CH_3_	0.910
**2i**	O	H	CH_3_	H	CH_3_	0.904
**2h**	O	H	OCH_3_	H	CH_3_	0.900
**1d**	S	OCH_3_	OCH_3_	OCH_3_	H	0.900
**1c**	S	H	CH_3_	H	H	0.898
**1f**	S	Cl	F	H	H	0.890
**1g**	S	H	H	H	CH_3_	0.888
**1i**	S	H	CH_3_	H	CH_3_	0.882
**1j**	S	OCH_3_	OCH_3_	OCH_3_	CH_3_	0.882
**1l**	S	Cl	F	H	CH_3_	0.880
**2d**	O	OCH_3_	OCH_3_	OCH_3_	H	0.880
**1a**	S	H	H	H	H	0.880
**1h**	S	H	OCH_3_	H	CH_3_	0.880
**2k**	O	H	CF_3_	H	CH_3_	0.820
**1e**	S	H	CF_3_	H	H	0.820
**2j**	O	OCH_3_	OCH_3_	OCH_3_	CH_3_	0.862
**2e**	O	H	CF_3_	H	H	0.836
**1k**	S	H	CF_3_	H	CH_3_	0.800

**Table 2 ijms-24-08377-t002:** Drug-likeness parameters calculated for the selected 24 compounds.

Compound	Lipinski Violations	Ghose Violations	Veber Violations	Egan Violations	PAINS Alerts	Total
**1a**	0	0	0	0	0	0
**1b**	0	0	0	1	0	1
**1c**	0	0	0	0	0	0
**1d**	1	2	2	1	0	6
**1e**	0	2	0	1	0	3
**1f**	0	0	0	0	0	0
**1g**	0	0	0	0	0	0
**1h**	0	0	0	1	0	1
**1i**	0	0	0	0	0	0
**1j**	1	2	2	1	0	6
**1k**	1	2	0	1	0	4
**1l**	0	2	0	0	0	2
**2a**	0	0	0	0	0	0
**2b**	0	0	0	0	0	0
**2c**	0	0	0	0	0	0
**2d**	1	2	1	1	0	5
**2e**	0	1	0	0	0	1
**2f**	0	0	0	0	0	0
**2g**	0	0	0	0	0	0
**2h**	0	0	0	0	0	0
**2i**	0	0	0	0	0	0
**2j**	2	2	1	1	0	6
**2k**	0	2	0	1	0	3
**2l**	0	0	0	0	0	0

**Table 3 ijms-24-08377-t003:** IFD and docking output results for **1a**–**l** and **2a**–**l** and the co-crystallized ligand nirmatrelvir (pdb code 7VH8) [51].

SARS-CoV-2 M^PRO^ (pdb Code 7VH8)		
Title	IFD Score	Docking Score
**1d**	−675.768	−8.979
**2l**	−675.108	−12.040
**1j**	−674.838	−7.595
**1f**	−674.292	−9.781
**1i**	−674.180	−9.222
**2i**	−674.046	−11.050
**2h**	−674.040	−10.969
**1l**	−674.037	−7.673
**1a**	−673.969	−8.573
**1k**	−673.927	−10.744
**1c**	−673.740	−8.008
**1b**	−673.730	−8.314
nirmatrelvir	−673.142	−10.169
**2c**	−673.071	−10.733
**1h**	−673.014	−7.862
**2j**	−672.880	−9.567
**2a**	−672.879	−10.861
**1g**	−672.752	−8.145
**1e**	−672.547	−8.150
**2k**	−672.538	−10.328
**2g**	−672.284	−10.226
**2d**	−672.184	−9.634
**2b**	−671.756	−10.415
**2f**	−671.736	−10.352
**2e**	−671.460	−9.900

**Table 4 ijms-24-08377-t004:** Overview of the amino acids involved in the binding of the selected 12 compounds with IFD scores higher than nirmatrelvir at the SARS-CoV-2 M^PRO^ catalytic binding site in proximity of 3 Å.

SARS-CoV-2 M^PRO^ (pdb Code 7VH8)																									
		S1′						S1							S2							S3/S4			
Title	IFD Score	T25	T26	L27	H41	V42	C145	F140	L141	N142	G143	H163	E166	H172	M49	M165	L167	P168	V186	D187	R188	Q189	T190	Q192	TOT
**1d**	−675.768	X	X	X	X	X	X	X	X	X		X	XX			X				X	X	X			16
**2l**	−675.108	X		X	X	X	X	X	X	X		X	X			X	X	X	X	X	X	X	X	X	19
**1j**	−674.838	X	X	X	X		X			X	X		XX		X	X				X	X	X			14
**1f**	−674.292	X	X	X	X	X	X	X		X		X	X		X	X				X	X	X			16
**1i**	−674.180	X	X	X			X	X	X	X	X	X	X	X	X	X	X	X	X		X	XX	X	X	21
**2i**	−674.046	X	X	X	X		X	X	X	X	X	X	X		X	X	X	X	X		X	X	X	X	20
**2h**	−674.040	X	X	X	X		X	X	X	X	X		X		X	X	X	X			X	X	X	X	18
**1l**	−674.037	X	X				X	X	X	X	X	X	X	X	X	X	X	X			X	X	X	X	18
**1a**	−673.969	X	X	X			X	X	X	X	X	X	X	X	X	X		X			X	X	X		17
**1k**	−673.927	X	X	X	X		X	X	X	X	X		X			X	X	X			X	X	X	X	17
**1c**	−673.740	X	X	X	X		X	X	X	X	X	X	X	X	X	X		X			X	X	X		18
**1b**	−673.730	X		X			X	X		X	X	X	X	X	X	X		X			X	X	X	X	17
nirmatrelvir	−673.142					X	X			X	X	X	X		X	X	X	X		X	X	X	X	X	15

**Table 5 ijms-24-08377-t005:** Distances calculated for each input molecules from pelitinib.

Title	PC1	PC2	Distance
pelitinib	−0.62	4.23	-
**1h**	−1.70	3.92	1.15
**2g**	−0.98	2.56	1.76
**2i**	0.70	2.91	1.90
**2c**	−0.67	1.76	2.53
**1i**	−2.98	5.49	2.66
**1b**	−3.08	2.73	2.92
**1c**	−4.63	4.16	4.02
**2a**	−2.85	0.91	4.05
**2h**	2.46	1.59	4.09
**1g**	−4.78	4.90	4.22
**1l**	−3.35	0.69	4.53
**2b**	0.87	0.01	4.53
**2l**	−0.06	−1.55	5.86

**Table 6 ijms-24-08377-t006:** IFD and docking output results (pdb code 7AXM) [41].

Title	IFD Score	Docking Score
**1c**	−693.48	−7.005
**1b**	−692.66	−7.214
**2l**	−692.66	−6.327
**1l**	−692.59	−6.72
**2i**	−692.24	−7.522
**1g**	−692.05	−6.368
**1i**	−691.57	−5.981
**2b**	−691.36	−6.187
pelitinib	−691.09	−6.192
**2c**	−691.03	−6.675
**1h**	−690.98	−5.082
**2g**	−690.73	−5.954
**2h**	−690.62	−5.679
**2a**	−689.92	−6.238

**Table 7 ijms-24-08377-t007:** Overview of the amino acids involved in the binding with the selected 13 compounds at the SARS-CoV-2 M^PRO^ allosteric site in proximity of 3 Å.

SARS-CoV-2 M^PRO^ (pdb Code 7AXM)																									
Title	IFD Score	S1	G2	D153	Y154	T209	A210	I213	N214	I249	P252	L253	A255	Q256	F294	V296	V297	R298	C300	S301	G302	V303	T304	F305	TOT
**1c**	−693.48			X	X						X			X			X	X	X	X		X	X		10
**1b**	−692.66	X	X					X	X		X	X		X			X		XX	X		X			12
**2l**	−692.66				X					X	X			X	X		X	X		XX	X	X	X		12
**1l**	−692.59			X	X						X		X	X			X	X	X			X	X	X	11
**2i**	−692.24					X	X	X			X	X	X	X		X	X		X	XX	X	X	X		15
**1g**	−692.05			X	X						X			X	X		X	X		X		X	X	X	11
**2h**	−691.57				X										X		X	X		X	X	X	X		8
**1i**	−691.36					X	X	X				X		X		X	X		X	X	X	X			11
**2b**	−691.09				X			X			X		X	X			X	X	X	X	X	X	X		12
pelitinib	−693.48										X	X		X		X	X		X	X		X			8

**Table 8 ijms-24-08377-t008:** Parameters docking combinations for RMDS value optimization.

Radii Van der Waals Scaling			Side Chain Optimization	Energy Minimization		RMSD	
Receptor Van der Waals Scaling	Ligand Van der Waals Scaling	Partial Charge Cut-Off	Residue Refinement	Distance-Dependent Dielectric Constant	Maximum Number of Minimization Steps	pdb Code 7VH8	pdb Code 7AXM
1.50	1.50	0.75	3 Å	0.5	20	0.87 Å	0.86 Å
1.25	1.25	0.50	3.5 Å	0.75	40	0.73 Å	0.75 Å
1.00	1.00	0.35	4 Å	1.00	60	0.66 Å	0.68 Å
0.75	0.75	0.25	4.5 Å	1.50	80	0.59 Å	0.57 Å
0.50	0.50	0.15	5 Å	2.0	100	0.51 Å	0.51 Å

## Data Availability

The data involved in this study are available within this article and in Appendix A.

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
