# Peer review of "In Silico Design of New Dual Inhibitors of SARS-CoV-2 MPRO through Ligand- and Structure-Based Methods"

_ijms, 2023, doi:10.3390/ijms24098377_

Round 1

Reviewer 1 Report

Bono et al focused on identifying potential inhibitors of the SARS-CoV-2 main protease that could prevent both dimerization and proteolytic activity, which could be useful in arresting the replication process of the virus. The methods used to identify the inhibitors are briefly described, including the use of a hybrid virtual screening protocol and the Biotarget Predictor Tool to filter a structural database. The ADME properties of the identified compounds were investigated, and Induced Fit Docking studies were performed to confirm predictions. Principal Component Analysis and docking protocol at the SARS-CoV-2 MPRO dimerization site were used to identify promising drug molecules. I would like some suggestions to consider in the revised version.

1.       How the proposed inhibitor is stable in the binding pocket.

2.       Authors need to discuss the merits and limitations of the study.

Author Response

We would like to thank you Reviewer 1 for the positive comments and the constructive analysis of our manuscript.

We have followed, point-by-point, your suggestions, and we hope that the revised version could be reconsidered for publication in the International Journal of Molecular Sciences.

You can find below the details of the revision steps:

  1. Answer: How the proposed inhibitor is stable in the binding pocket.

Response: to evaluate the stability of the ligand into the target binding site, molecular dynamic simulations were performed for the two representative compounds 1c and 1d (into the catalytic domain and the dimerization domain, respectively). Both complexes demonstrated high stability across the simulation time (the results are reported in the text in paragraph).

  1. Answer: Authors need to discuss the merits and limitations of the study.

Response: in the conclusion section we added a short discussion about the major merits and limitations of the proposed VS in comparison with the other VSs focused on SARS-CoV-2 MPRO reported in the literature in the past three years. In detail, the main novelty of our approach was the integration of an in-house molecular descriptor-based tool (BIOTARGET finder) with classical structure-based techniques (molecular docking and molecular dynamic) and interestingly, statistical multivariate analysis. The proposed chemometric protocol, as reported in the results, demonstrated high reliability and robustness in the virtual screening of huge databases of structures. Furthermore, to the best of our knowledge, this is one of the first examples of VSs focused on the identification of dual SARS-CoV-2 MPRO inhibitors.

Reviewer 2 Report

In this work, Bono et al. performed a structure-based virtual screening (VS) targeting the SARS-CoV-2 Main Protease (MPRO), aimed at finding compounds capable of interacting with both the active site and the allosteric one involved in the protein dimerization.

The VS workflow, focused on an in-house database of heterocyclic compounds, is overall well-designed, but it lacks important cross-checks/validation steps.

Considering the flourishing literature on the topic, and since numerous VS works were published during the early-to-mid phase of the pandemic, it is my opinion that these aspects should be significantly improved before publication. In the last three years, many studies have been retracted due to inconsistencies and/or flawed design. At a certain point, computational-only papers were not even accepted in some archive repositories. 

In addition, the authors are disregarding results from previous VS campaigns on this target and do not clarify how different is their outcome from previously published works. At the very least, the most notable results from previous studies on MPRO should be mentioned in the introduction. Drugs whose usefulness in the treatment of SARS-CoV-2 has been disproved should not be mentioned in the introduction as potential therapeutics for COVID-19 (e.g.: lopinavir/ritonavir, chloroquine/hydroxychloroquine).

In more detail:

1. The "Methods" section does not provide any info, or very little, about the definition of the docking grid and optimization of the docking parameters. It is suggested to improve this section.

2. It is strongly suggested to report the results of redocking calculations performed using the chosen X-ray structures (7VH8 and 7AXM) and the identified docking grid/parameters. If the authors did not do so, it is strongly suggested to perform them, as they could provide a good indication of the quality of these parameters, or could provide insights about how to improve them.

3. The rationale behind the choice of the threshold for the PC distance from pelitinib is not explained. How it was chosen?

4. Docking calculations are lacking validation. It is strongly suggested to perform molecular dynamics calculations on the docked poses to provide further assessment of their stability.

5. It would be useful to compare the outcome of the work with the findings of previous VS studies targeting MPRO, to better contextualize the novelty of the paper.

Author Response

We would like to thank you Reviewer 2 for the constructive analysis of our manuscript.

We have followed, point-by-point, your suggestions, and we hope that the revised version could be reconsidered for publication in the International Journal of Molecular Sciences.

You can find below the responses:

In this work, Bono et al. performed a structure-based virtual screening (VS) targeting the SARS-CoV-2 Main Protease (MPRO), aimed at finding compounds capable of interacting with both the active site and the allosteric one involved in the protein dimerization.

The VS workflow, focused on an in-house database of heterocyclic compounds, is overall well-designed, but it lacks important cross-checks/validation steps.

Considering the flourishing literature on the topic, and since numerous VS works were published during the early-to-mid phase of the pandemic, it is my opinion that these aspects should be significantly improved before publication. In the last three years, many studies have been retracted due to inconsistencies and/or flawed design. At a certain point, computational-only papers were not even accepted in some archive repositories. 

In addition, the authors are disregarding results from previous VS campaigns on this target and do not clarify how different is their outcome from previously published works. At the very least, the most notable results from previous studies on MPRO should be mentioned in the introduction. Drugs whose usefulness in the treatment of SARS-CoV-2 has been disproved should not be mentioned in the introduction as potential therapeutics for COVID-19 (e.g.: lopinavir/ritonavir, chloroquine/hydroxychloroquine).

Response: In the introduction section we added a short overview of the in-silico techniques/approaches adopted in the VSs focused on SARS-CoV-2 MPRO and described in literature to date. In addition, the discussion of the potential therapeutics for COVID-19 whose usefulness has been disproved (lopinavir/ritonavir, chloroquine/hydroxychloroquine etc..), as suggested, was removed.

  1. The "Methods" section does not provide any info, or very little, about the definition of the docking grid and optimization of the docking parameters. It is suggested to improve this section.

Response: A detailed description of all the parameters selected for the docking studies has been added in the Methods section (3.2.3. Docking validation, Table 8).

  1. It is strongly suggested to report the results of redocking calculations performed using the chosen X-ray structures (7VH8 and 7AXM) and the identified docking grid/parameters. If the authors did not do so, it is strongly suggested to perform them, as they could provide a good indication of the quality of these parameters or could provide insights about how to improve them.

Response: the results of redocking calculations have been added in the methods section, confirming quite good RMSD values for the original ligands (nirmatrelvir and pelitinib for 7VH8 and 7AXM, respectively).

  1. The rationale behind the choice of the threshold for the PC distance from pelitinib is not explained. How was it chosen?

Response: the Grubb’s test was performed to identify the outliers in the statistical analysis. In detail, as described in section 3.3, the outliers were determined by evaluating singularly those compounds outside the red circle (Figure 8) in comparison with the cluster of molecules closest to pelitinib. The identified outliers were not included in the next step of the virtual screening.

  1. Docking calculations are lacking validation. It is strongly suggested to perform molecular dynamics calculations on the docked poses to provide further assessment of their stability.

Response: molecular dynamic calculations were performed on the best-docked pose of 1d and 1c, identified as representative compounds for the catalytic and dimerization site, respectively. Both complexes demonstrated high stability across the simulation time, confirming the robustness of the adopted in silico protocol (the results are included in the text in paragraph).

  1. It would be useful to compare the outcome of the work with the findings of previous VS studies targeting MPRO, to better contextualize the novelty of the paper.

Response: in the conclusion section we added a short discussion about the major merits and limitations of our VS in comparison with the other VSs focused on SARS-CoV-2 MPRO reported in literature in the past three years. In detail, the main novelty of our approach was the integration of an in-house molecular descriptor-based tool (BIOTARGET finder) with classical structure-based techniques (docking and molecular dynamic) and interestingly, statistical multivariate analysis. The proposed chemometric protocol, as reported in the results, demonstrated high reliability and robustness in the virtual screening of huge databases of structures. Furthermore, to the best of our knowledge, this is one of the first examples of VSs focused on the identification of dual SARS-CoV-2 MPRO inhibitors.